# Semiparametric mixed-effects model for analysis of non-invasive longitudinal hemodynamic responses during bone graft healing

**Sami Leon**[1⊙], **Jingxuan Ren**[2⊙], **Regine Choe**[2,3], **Tong Tong Wu**[1]*

**1** Department of Biostatistics and Computational Biology, University of Rochester, Rochester, NY, United States of America, **2** Department of Biomedical Engineering, University of Rochester, Rochester, NY, United States of America, **3** Department of Electrical and Computer Engineering, University of Rochester, Rochester, NY, United States of America

⊙ These authors contributed equally to this work.
* Tongtong_Wu@urmc.rochester.edu

**Data Availability Statement:** All relevant data are within the paper and its Supporting information files.

## Abstract

When dealing with longitudinal data, linear mixed-effects models (LMMs) are often used by researchers. However, LMMs are not always the most adequate models, especially if we expect a nonlinear relationship between the outcome and a continuous covariate. To allow for more flexibility, we propose the use of a semiparametric mixed-effects model to evaluate the overall treatment effect on the hemodynamic responses during bone graft healing and build a prediction model for the healing process. The model relies on a closed-form expectation–maximization algorithm, where the unknown nonlinear function is estimated using a Lasso-type procedure. Using this model, we were able to estimate the effect of time for individual mice in each group in a nonparametric fashion and the effect of the treatment while accounting for correlation between observations due to the repeated measurements. The treatment effect was found to be statistically significant, with the autograft group having higher total hemoglobin concentration than the allograft group.

## Introduction

Critical-sized bone defects are those that cannot heal without intervention. Each year, more than 2.2 million bone grafting procedures are performed worldwide for treating critical-sized bone defects [1, 2]. The gold standard treatment is to implant allograft harvested from cadavers to the defect site. However, such allograft is devitalized to prevent immune response therefore leading to a high long-term failure rate [3, 4]. Animal models including mice are widely used for developing and evaluating new treatments to enhance allograft healing. For example, a hydrogel-based tissue-engineered periosteum delivering stem cells to the allograft was found to enhance allograft healing in a mouse model [5, 6]. Vascularization, the development of blood vessels, is widely measured for evaluating the bone healing in these studies [7–9], since

**Funding:** RC and TTW's work was partly supported by grant from the National Institutes of Health NIH R01AR071363; TTW's work was also supported by grant from the National Science Foundation NSF CCF-1934962. The funders had no role in study design, data collection and analysis, decision to publish, or preparation of the manuscript.

**Competing interests:** The authors have declared that no competing interests exist.

vascularization is regarded as a precursor of bone formation [8, 10, 11]. In animal studies, vasculatures in the bone are measured using contrast agent-mediated micro-computed tomography (micro-CT) [5, 6, 12]. This procedure is terminal thus making the longitudinal vascularization monitoring of an individual mouse impossible. Therefore, a non-terminal *in vivo* technique is needed for frequent longitudinal monitoring of the bone vascularization.

Previously, we applied spatial frequency domain imaging (SFDI) for measuring the amount and the quality of vascularization, in particular, total hemoglobin concentration (*THC*) and blood oxygen saturation (*StO$_2$*) in a mouse femoral model [13]. SFDI is a non-invasive imaging technique where near-infrared (NIR) light is employed to quantify *THC* and *StO$_2$* [14, 15]. In that study, mice with two types of femoral grafts, autograft and allograft, were measured longitudinally from one day before injury to day 44 post-injury. Different from allograft, autograft is harvested from the same injured mouse without devitalization. Therefore, autograft is regarded to have a better vascularization and healing potential [5, 6, 16–18]. However, the detailed temporal trend of the vascularization was not readily available, partially due to the lack of noninvasive monitoring tools. To find out the optimal time points for monitoring, we first performed daily SFDI measurement in the first two weeks. The measurement frequency decreased to twice a week after week 2 based on the observation that mice started to develop resistance to anesthesia. When analyzing the data, we excluded the data in the first week due to signal contamination caused by the sutures covering the wound site. All these adaptations in measurement frequency and data analysis resulted in uneven time points in the longitudinal data.

To account for the repeated measurements from the same mouse in the longitudinal monitoring, the data were analyzed using a linear mixed-effects model (LMM). LMMs are used to describe the relationship between the response (e.g., *THC* or *StO$_2$*) and a set of predictors (e.g., time) that are clustered according to one or more classification factors, and hence are ideal to analyze repeated measures data. They assume a linear relationship between the response and the predictors, which, however, might over-simplify the relationship. In our study, the physiological changes of *THC* or *StO$_2$* during the bone healing are usually nonlinear. Therefore, in the previous analysis [13], the time indicator was treated as a categorical variable to give more flexibility. By fitting a LMM, a significant treatment effect was detected for *THC* between the two graft types [13]. However, this approach has several limitations. First, the linearity assumption is too stringent and not desirable in many situations as people might expect a nonlinear relationship between the response and predictors. Second, by treating the time indicator as a categorical variable, the LMM allows an arbitrary pattern of temporal changes. However, it is unable to make predictions at a time point different than those categorical time points (e.g., one cannot make prediction at day 2 or 3, which is not part of the time points when data were collected). Moreover, a LMM is not suitable for datasets with a large number of time points when the number of subjects is small [19], e.g., time series data. Given these reasons, we need a mixed-effects model that can estimate both treatment effect and the nonlinear pattern of temporal effects.

In this study, we propose the use of a semiparametric mixed-effects model (SMM) [20, 21], which is the the state of art in longitudinal data modeling, for the analysis of the longitudinal vascularization data during mouse femoral graft healing described in [13]. Our aim is to evaluate the overall treatment effect and build a prediction model for the healing process. By fitting a SMM, we are able to (1) test if there exists a significant overall treatment effect; (2) estimate the effect size of treatment; (3) model the relationship between the total hemoglobin concentration and treatment over time; (4) model the nonlinear pattern of the temporal effects; (5) make prediction at any arbitrary time point during the healing process. An additional advantage of SMM is that varying time points are allowed, e.g., due to adaptation made to

accommodate the development of anesthesia resistance, etc. Last but not least, a SMM could also enable researchers to identify the earliest time point that begins to show a group difference. As discussed later in the Results section, we could choose Day 7 for *rTHC* and be confident the overall treatment effects would be sustained throughout the following time period.

In SMM, the longitudinal response can be considered as a function of time, where time is treated as a continuous variable. The model fitting is more data-driven and has no restriction on the shape of the fitted model. By doing that, a nonlinear relationship is allowed between the longitudinal response and predictors. The second advantage is that prediction can be made at any time since time is continuous. Third, the model works well in the situation where there are a large number of time points with only a few observations or even only one observation at a time.

The rest of the paper is organized as follows. We will describe the data and the SMM in the next section. The results using SMM will be then presented. The paper will be concluded with a discussion at the end.

## Materials and methods

### Data description

SFDI data from our previous study [13] are presented in Fig 1. The hemodynamic observations (*THC* and *StO₂*) of each group from the longitudinal SFDI measurements are shown. Graft *THC* and *StO₂* were extracted from two-dimensional SFDI images of the mouse hindlimb at each day, by taking the mean within a rectangular region of interest (ROI). The ROI was defined over the mid-diaphysis, which is the location of the implanted graft. The hemodynamic observations of the allografts and autografts are shown on the left and right column of Fig 1, respectively. Qualitatively speaking, the autograft group exhibits higher *THC* than the allograft group in the later stage of the healing (e.g., approximately after 3 weeks). The $StO_2$ of both groups fluctuates around the value of 40% during the healing process.

### Semiparametric mixed-effects model

To account for the repeated measurements of each mouse, a SMM was used to model the longitudinal response variables with random intercepts. We used the total hemoglobin concentration (*THC*) as the example in this section, and similar models could be fitted for other response variables. The SMM is defined as follows:

$$THC = \beta_0 + \beta_1 g + f_g(t) + \phi + \epsilon,$$

where $t$ is the time (in days), $g$ is the graft type (0 for allograft and 1 for autograft), $\phi$ is the random intercept for individual subjects, and $\epsilon$ is the random noise. The intercept $\beta_0$ indicates the overall mean of *THC*, $\beta_1$ is the regression coefficients of $g$, and $f_g(t)$ belongs to the space of polynomials and represents the nonparametric part of the model which captures the effect of time for each mouse in each group. The function $f_g$ is unknown and needs to be estimated.

As a comparison, the LMM is given by:

$$THC = \beta_0 + \beta_1 g + \beta_2 t + \beta_3 t \cdot g + \phi + \epsilon,$$

where the time related part $\beta_2 t + \beta_3 t \cdot g = (\beta_2 + \beta_3 g) \cdot t$ is the counterpart of $f_g(t)$ in SMM. It is easy to see that LMM assumes a linear time trend, i.e., $(\beta_2 + \beta_3 g) \cdot t$, while $f_g(t)$ is the nonlinear function of time in SMM. In other words, the SMM models the joint effects of time and group through $f_g(t)$, the LMM captures the effects using two linear terms $\beta_2 t + \beta_3 t \cdot g$. The latter has some limitations that we can circumvent using a SMM, as discussed in the introduction.

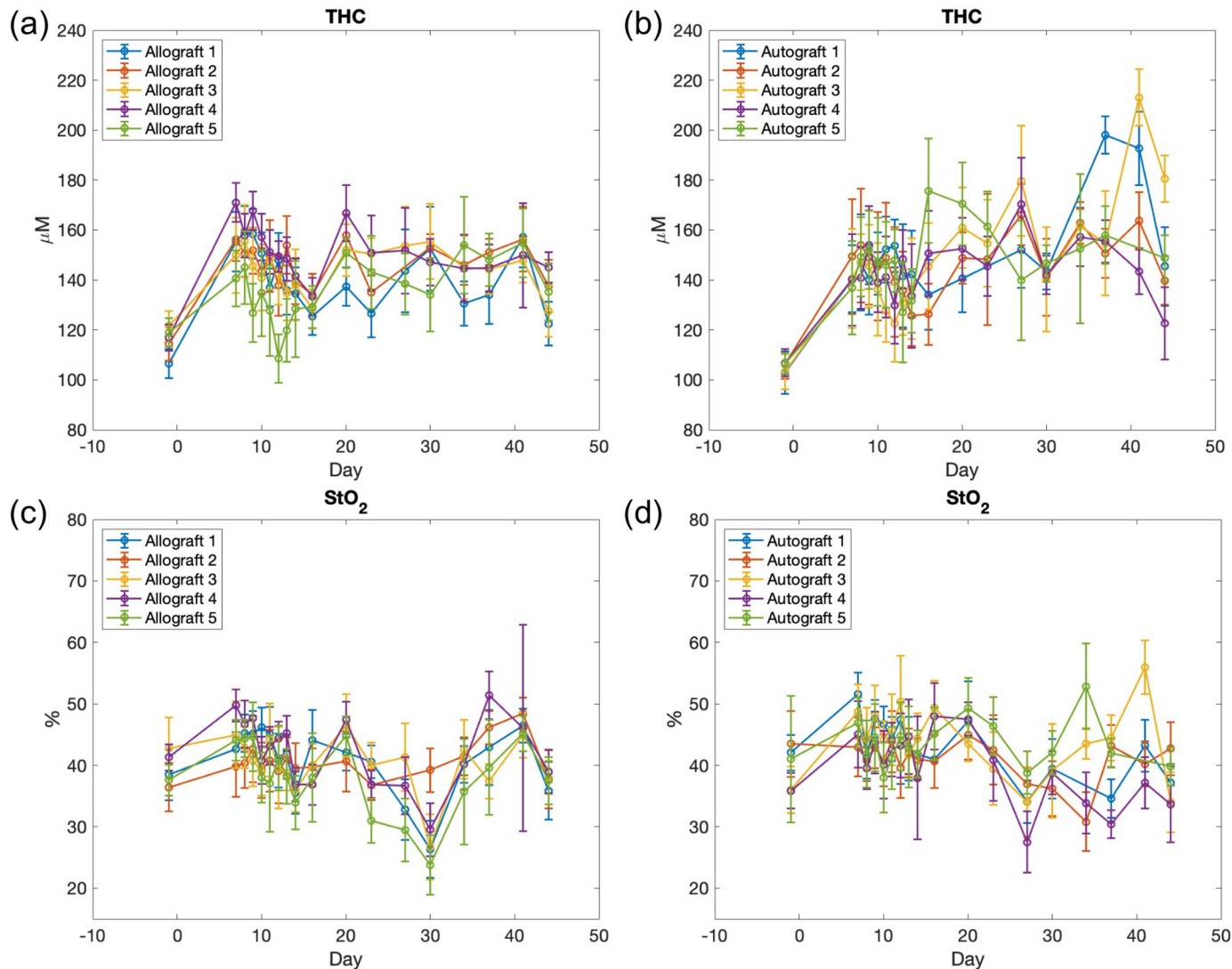

**Fig 1. Temporal changes of graft hemodynamic parameters in individual mouse are grouped based on parameter and graft.** (a) *THC* of the allograft group; (b) *THC* of the autograft group; (c) *StO₂* of the allograft group; (d) *StO₂* of the autograft group. The error bar indicates the standard deviation of the quantity (*THC* or *StO₂*) within the region of interest.

## Estimation of SMM

The parameters of the SMM are estimated using a penalized expectation-maximization (EM) algorithm proposed by [22], where at each iteration of the EM algorithm the parameters are estimated jointly. A penalty term is used to estimate a Lasso (least absolute shrinkage and selection operator)-type estimator [23] of $f_g$, the estimate of $\phi$ is obtained in the M-step of the algorithm, and the coefficients $\beta_0$ and $\beta_1$ are estimated using least-squares based on the estimates of $f_g = F\lambda$ and $\phi$.

Let $N$ be the number of mice with $n_i$ observations each, and $n = \sum_{i=1}^{N} n_i$. A set $\{\psi_1, \ldots, \psi_M\}$ of potential basis functions is provided to the algorithm, and the Lasso procedure will

automatically select the basis functions. At iteration $h + 1$, $\lambda$ can be updated as

$$\lambda^{(h+1)} = \arg\min_{\lambda} \|THC - \beta_0^{(h)} - \beta_1^{(h)}g - F\lambda - \phi^{(h+1)}\|_n^2 + 2\sigma^{2,(h+1)}\sum_{k=1}^{2M} r_{n,k}|\lambda_k|,$$

where

$$F = \begin{pmatrix} F_0 & 0 \\ 0 & F_1 \end{pmatrix}$$

is a block matrix, $F_0 = (\psi_k(t_{ij}))_{k,i,j}$ with $t_{ij}$ only corresponding to times of mouse receiving allograft, similarly $F_1 = (\psi_k(t_{ij}))_{k,i,j}$ with $t_{ij}$ only corresponding to times of mouse receiving autograft, $r_{n,k} = \|\psi_k\|_n \sqrt{\gamma \ \log \ 2M/n}$ is a penalty term, $\gamma > 0$ is the tuning parameter and $\|\psi_k\|_n^2 = \frac{1}{n}\sum_{i=1}^{N}\sum_{j=1}^{n_i}\psi_k^2(t_{ij})$.

The function $f_g$ is finally estimated as

$$f_g^{(h+1)} = \begin{cases} \displaystyle\sum_{k=1}^{M}\lambda_k^{(h+1)}\psi_k & \text{if } g = 0, \\[2em] \displaystyle\sum_{k=1}^{M}\lambda_{M+k}^{(h+1)}\psi_k & \text{if } g = 1. \end{cases}$$

The tuning parameter $\gamma$ controls the smoothness of $f_g$, where a larger value of $\gamma$ will induce more $\lambda$ coefficients to be set to zero, which will result in a smoother curve. In our analysis, the set of basis functions is given by cubic splines, Fourier bases, power functions and Haar functions.

The two parameters $\phi^{(h+1)}$ and $\sigma^{2,(h+1)}$ are estimated following the two steps of the EM algorithm:

- *E-step*:

$$\begin{cases} \phi^{(h+1)} = \dfrac{W^{(h)}}{\sigma^{2,(h)}}Z'(THC - F\lambda^{(h)}) \\[1.5em] W^{(h+1)} = \sigma^{2,(h)}(Z'Z + \eta^{(h)})^{-1} \end{cases}$$

where $Z = \text{diag}(Z_1, \ldots, Z_N)$ with $Z_i$ a vector of $n_i$ ones, $\eta^{(h)} = \sigma^{2,(h)}/\sigma_\phi^{2,(h)}$ and W is the conditional variance of $\phi$.

- *M-step*:

$$\begin{cases} \sigma_\phi^{2,(h+1)} = \dfrac{1}{N}\left[\displaystyle\sum_{i=1}^{N}\phi_i^{2,(h+1)} + \dfrac{\sigma^{2,(h)}}{n_i + \eta^{(h)}}\right] \\[2em] \sigma^{2,(h+1)} = \dfrac{1}{n}\left[\displaystyle\sum_{i=1}^{N}\sum_{j=1}^{n_i}\hat{E}_{ij}'\hat{E}_{ij} + \sum_{i=1}^{N}\dfrac{n_i\sigma^{2,(h)}}{n_i + \eta^{(h)}}\right] \end{cases}$$

where $\hat{E} = Y - \phi^{(h+1)} - F\lambda^{(h)}$.

The fixed effects $\beta = (\beta_0, \beta_1)'$ are estimated using least-squares applied on $THC - F\lambda^{(h+1)} - \phi^{(h+1)}$, by solving $X'X\beta^{(h+1)} = X'(THC - F\lambda^{(h+1)} - \phi^{(h+1)})$, with X the design matrix with first column a vector of ones, and second column g.

We refer to [22] for details on how to derive the formulas. Some considerations for the tuning parameter $\gamma$ are given in [24], and the authors proved that for $\gamma > 2$ the estimator of $f_g$ satisfies an oracle inequality. However, since $\gamma$ also influences the stability of the EM algorithm, their recommendation is then to choose a $\gamma$ value that is close to 2, which makes the EM algorithm more stable.

## Results

### Choice of $\gamma$

**Effects of $\gamma$ on the significance of $\beta$.**    The model estimation depends on the value of the tuning parameter $\gamma$. We therefore examined the effects of $\gamma$ by plotting the coefficient $\beta_1$ of the group effect $g$ vs. $\gamma$ on a grid between 0 and 2, with increments of 0.05, shown in Fig 2 for *THC* and in Fig 3 for *rTHC*. In both figures, the choice of $\gamma$ has an impact on the value and significance of $\hat{\beta}_1$. For *THC* (Fig 2), when $\gamma = 0$ or 0.05, the estimated group effect coefficient is insignificant ($p$-value $> 0.05$); while $\gamma \geq 0.1$ makes the estimated group coefficient significant. For *rTHC* (Fig 3), the $\gamma$ value does not affect the statistical significance of the group coefficient—as long as $\gamma > 0$, $\hat{\beta}_1$ is always significant with $p < 0.05$. In both cases, the estimated regression coefficient is zero when $\gamma = 0$. However, $\gamma = 0$ is not a reasonable choice since the estimate of $f_g(t)$ will be the group mean at each time point, and there is then no information left to be explained by the group coefficient.

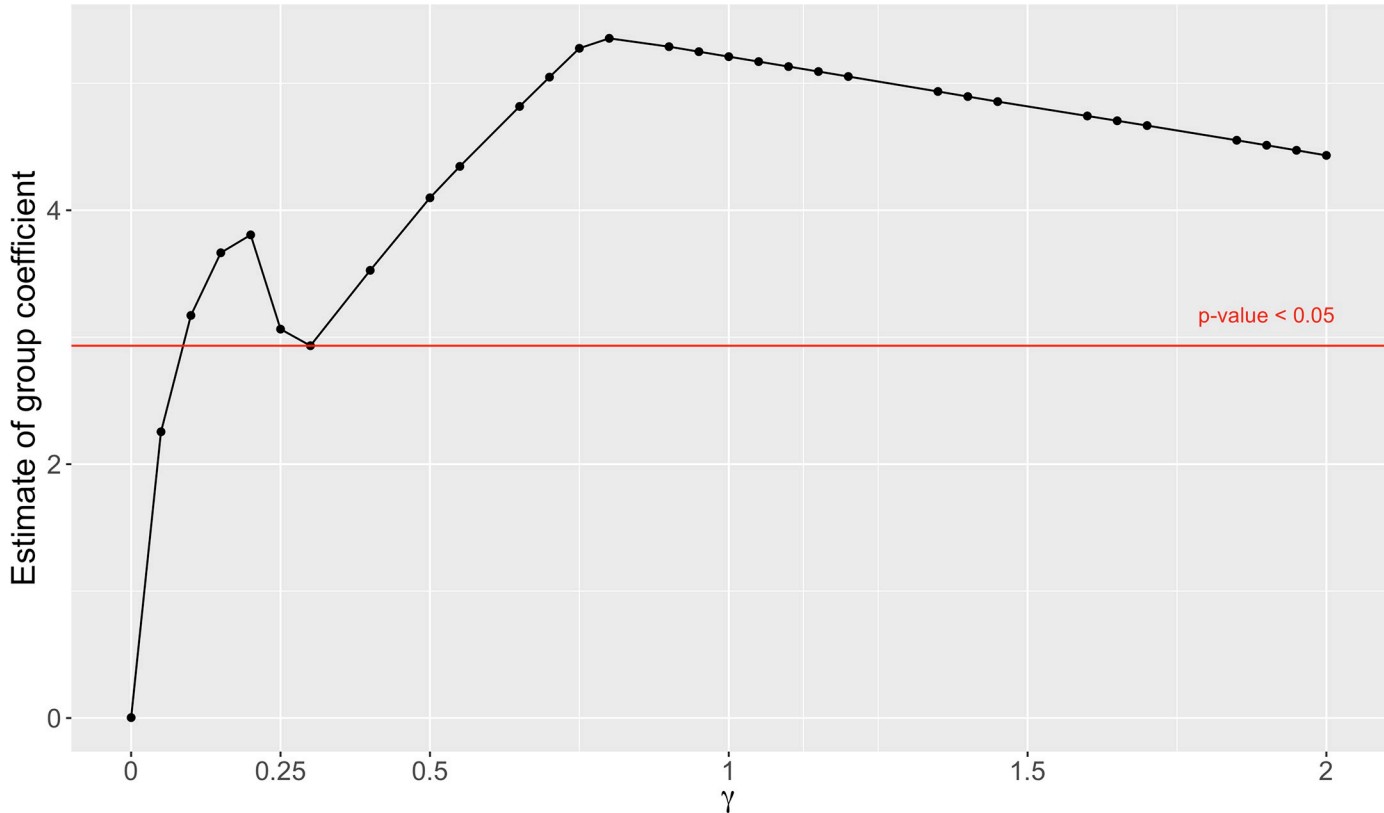

**Fig 2. Estimated group coefficient $\beta_1$ vs. $\gamma$ for *THC*.** The group coefficient $\beta_1$ is estimated at different $\gamma$ values. Any coefficients above the red line are statistically significant with $p < 0.05$, the red line crosses the curve at the point $(\gamma = 0.3, \hat{\beta}_1 = 2.9)$.

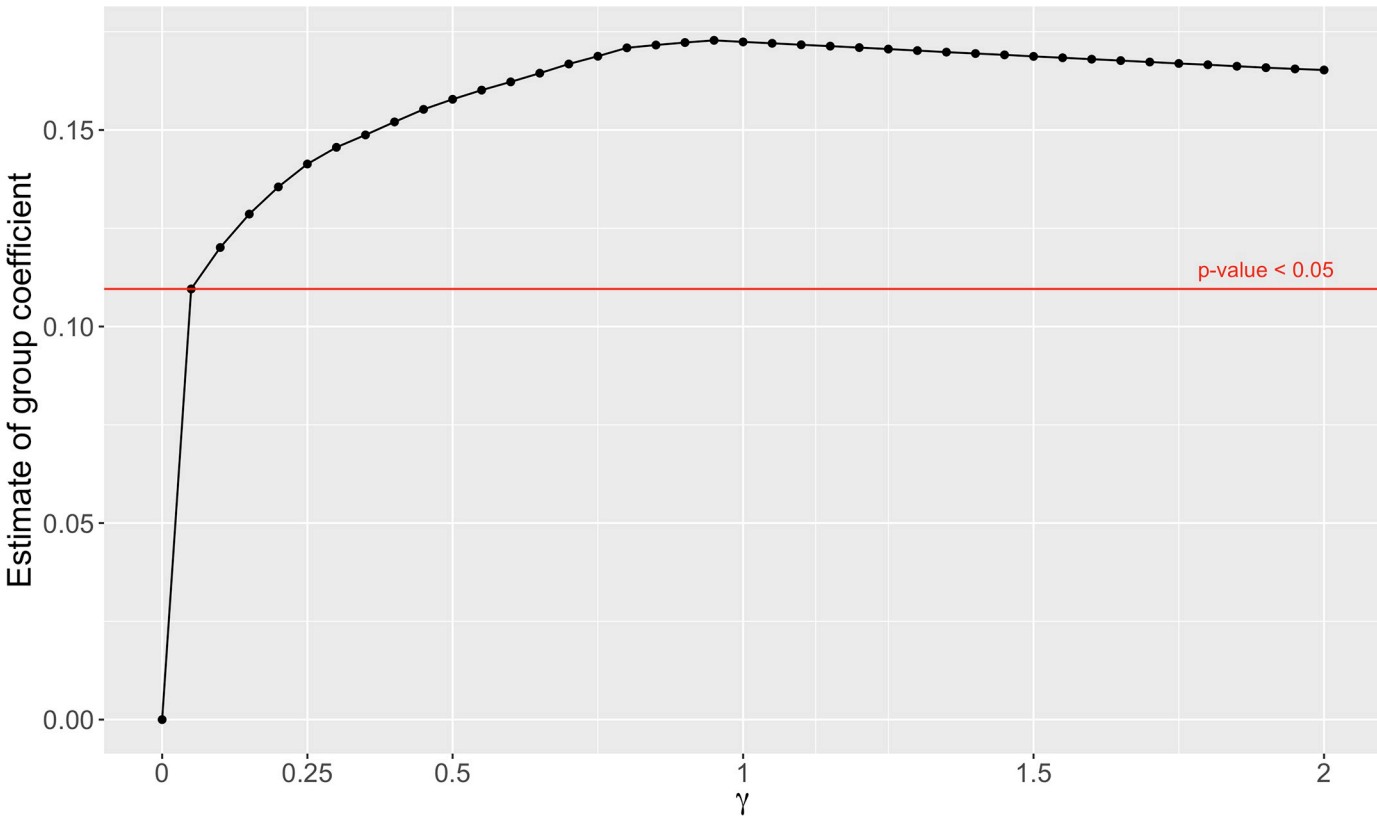

**Fig 3. Estimated group coefficient $\beta_1$ vs. $\gamma$ for $rTHC$.** The group coefficient $\beta_1$ is estimated at different $\gamma$ values. Any coefficients above the red line are statistically significant, the red line crosses the curve at the point ($\gamma = 0.05, \hat{\beta}_1 = 0.11$).

We observed similar patterns for the response variable $StO_2$. The choice of $\gamma$ has an impact on the value of regression coefficients, but does not affect much on the statistical significance. We therefore omit the plots for $StO_2$ and $rStO_2$.

**Effects of $\gamma$ on the smoothness of $f_g(t)$.**    In Fig 4 we plotted the estimated $THC$ trajectories for both treatment groups using a SMM with different values of $\gamma$. There is a clear trade-off between fitting the data adequately and getting a smooth curve. We can see that $\gamma = 0$ and $\gamma = 2$ do not appear to be good candidates, since one follows every small variation of the data and the other one is almost a straight line. We then chose a value of $\gamma$ that was able to capture the variations of the data while remaining smooth, our choice is $\gamma = 0.5$. For the rest of the analysis the results will be based on this value.

### Results for $THC$

**THC.**    The allograft group and autograft group appear to have different trends. They both show an initial increase, but after this initial increase only the autograft group keeps on increasing, while the allograft group seems to roughly fluctuate around the same value of 145 $\mu M$. A formal test for a difference between the two groups can be performed during the least-squares step of the algorithm. The results are presented in Table 1, the intercept corresponds to the estimated $THC$ mean of the allograft group after subtracting the effect of time in each group, and the effect associated to the repeated measurements of each mouse. The group effect is statistically significant when using $\gamma = 0.5$, however some choices of $\gamma$ can make the

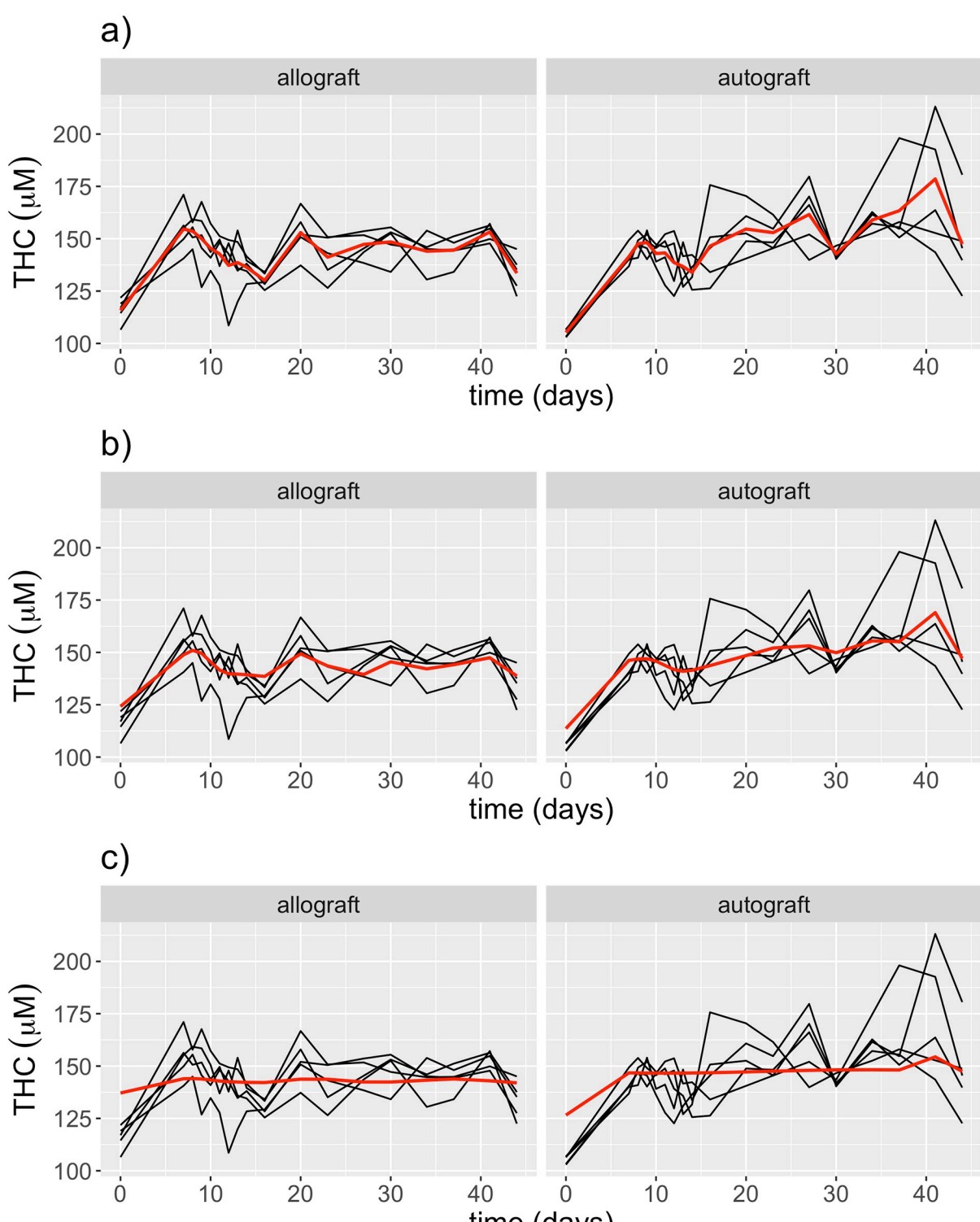

**Fig 4. Estimated *THC* by SMM (red curve) for the allograft group (left) and the autograft group (right) with different values of $\gamma$.** Black curves are observed *THC* trajectories. a) SMM estimates using $\gamma = 0$, b) SMM estimates using $\gamma = 0.5$, c) SMM estimates using $\gamma = 2$.

**Table 1. The estimate of group effect in the SMM for *THC* and *rTHC* at $\gamma$ = 0.5.**

| Outcome | Reg. Coeff. | Estimate | Std. Error | t value | Pr(> \|t\|) |
|---|---|---|---|---|---|
| *THC* | (Intercept) | 143.40 | 1.13 | 126.86 | $< 10^{-16}$ |
| | Group | 4.10 | 1.61 | 2.55 | 0.01 |
| *rTHC* | (Intercept) | 1.24 | 0.01 | 118.41 | $< 10^{-16}$ |
| | Group | 0.16 | 0.01 | 10.58 | $< 10^{-16}$ |

coefficient not statistically significant with a p-value close to 0.05 but slightly larger. It is then not clear if there is truly a difference between the two groups for *THC*. It can be seen from Fig 5(a) as well that the difference between the two estimated trajectories of the two treatments is not obvious until toward the end of the study.

 **rTHC.** From Fig 4, we can see that the pre-surgery *THC* value of the autograft group is lower than the allograft group. To account for the variation among individuals the same analysis is performed using a relative *THC* (*rTHC*), for each mouse the *THC* values are normalized by their pre-surgery *THC* values. The estimated *rTHC* trajectories for both groups using $\gamma$ = 0.5 are shown in Fig 5. From Fig 5, we see that the difference between the two groups is even more noticeable than when using *THC* and the difference begins as early as Day 7. As a result, the group effect is greater and highly significant. In Table 1, the estimated mean of the allograft group is 143.40 $\mu M$ and 147.50 $\mu M$ for the autograft group, which corresponds to a 2.86% increase. The effect from Table 1 suggests that the estimated relative mean of the autograft group is now 12.9% greater than the estimated relative mean of the allograft group.

## Results for $StO_2$

The same method is applied to the variable $StO_2$, which is the blood oxygen saturation of the graft and the surrounding soft tissues. In Fig 6, the groups appear to be similar in their trends, the main difference is at 30 days where the allograft group shows a larger decrease than the autograft group. Table 2 suggests that the difference is not statistically significant. The same

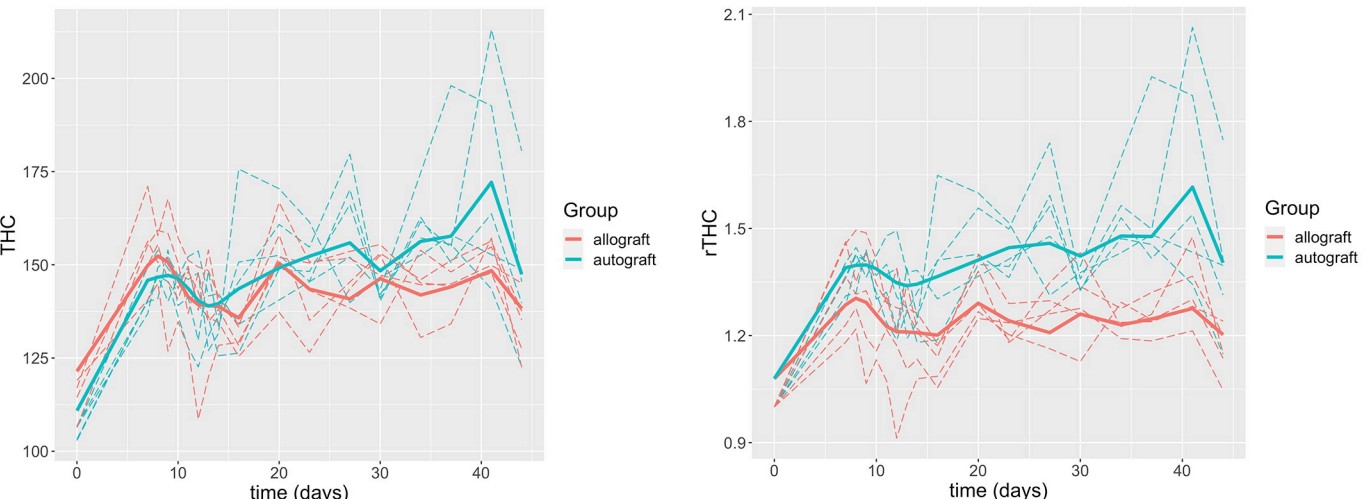

**Fig 5.** (a) Estimated *THC* trajectories over time by SMM (thick curves) for the allograft group (red) and the autograft group (blue) at $\gamma$ = 0.5. The dashed curves are the observed *THC* trajectories of individual mice in each group. (b) Estimated *rTHC* trajectories over time by SMM (thick curves) for the allograft group (red) and the autograft group (blue) at $\gamma$ = 0.5. The dashed curves are the observed *rTHC* trajectories of individual mice in each group.

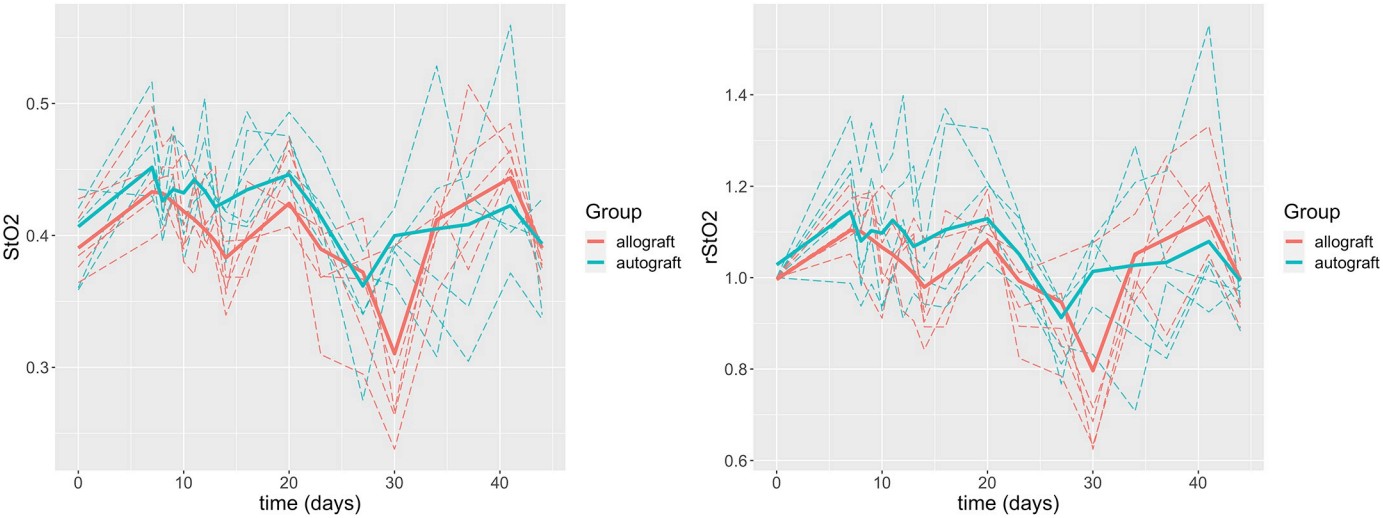

**Fig 6.** (a) Estimated $StO_2$ trajectories over time by SMM (thick curves) for the allograft group (red) and the autograft group (blue) at $\gamma = 0.5$. The dashed curves are the observed $StO_2$ trajectories of individual mice in each group. (b) Estimated $rStO_2$ trajectories over time by SMM (thick curves) for the allograft group (red) and the autograft group (blue) at $\gamma = 0.5$. The dashed curves are the observed $rStO_2$ trajectories of individual mice in each group.

analysis is performed on the relative $StO_2$ ($rStO_2$), using $\gamma = 0.5$. We see similar trends for both groups in Fig 6, and in Table 2 the difference is still not statistically significant.

## Discussion

In this study, we applied SMM for analyzing the hemodynamic data from a longitudinal mouse bone healing monitoring. Significant difference of the longitudinal $THC$ was found between the autograft and allograft groups. However, no significant difference of $StO_2$ was found between two groups. The result is consistent with that from the previous LMM method and agrees with the physiological expectation. Even though both the LMM and SMM indicated a significant treatment effect for this particular dataset, SMM is more versatile and advantageous than LMM. Instead of treating the time as a categorical variable, the SMM allows nonlinear relationships between the hemodynamic observation and the time. The continuous time in the model also allows prediction of the hemodynamic changes within the period of monitoring. In pre-clinical and clinical research, missing time points are common due to physiological, technical, or scheduling limitations. The SMM could help estimating the hemodynamic parameters at the missing time points.

One of the caveats of SMM is the complexity of the method, leading to difficulty in implementation. This may potentially impact the wide-spread usage of this particular method. However, the ability to handle non-linearity of the dataset in longitudinal analysis could greatly relax the stringent linearity assumption by the LMM method.

**Table 2. The estimate of group effect in the SMM for $StO_2$ and $rStO_2$ at $\gamma = 0.5$.**

| Outcome | Reg. Coeff. | Estimate | Std. Error | t value | Pr(> \|t\|) |
|---------|-------------|----------|------------|---------|-------------|
| $StO_2$ | (Intercept) | 0.41 | 0.004 | 106.02 | $< 2 \times 10^{-16}$ |
| | Group | 0.01 | 0.005 | 1.78 | 0.08 |
| $rStO_2$ | (Intercept) | 1.04 | 0.01 | 104.69 | $< 2 \times 10^{-16}$ |
| | Group | 0.02 | 0.01 | 1.41 | 0.16 |

Another limitation of SMM is the determination of the tuning parameter $\gamma$. In the paper of [22], the authors recommended some value larger than 2 but as close as 2. In our limited numerical experiments, we found such a value is not a good choice and leads to an overly smoothing result (see Fig 4). Our experience is to tune $\gamma$ over a grid between 0 and 2 and select a value that gives a balance of model fitting adequacy and smoothness. Of course, it is of great interest to develop a criterion that can be used to determine the optimal value automatically and objectively, which will be studied in the near future.

The third limitation of SMM is that it cannot account for the heteroscedasticity. It is shown in Fig 1(b) and 1(c) that the variation gets larger over time (the variation is fairly stable in (a) and (d)), while SMM assumes a constant variance, which is a limitation. We are not aware of any model that can take into account the heteroscedasticity while fulfilling all of our goals. Extending the current model by incorporating heteroscedasticity is out of the scope of this paper but will be considered in future research.

Last but not least, SMM is prone to overfitting due to the model complexity. The discovered nonlinear pattern could be the pattern of noises and might not be applicable outside the sample. Our findings would be tested using new observations in future studies.

In summary, the advantages of the SMM model makes it suitable for a broader range of biomedical applications. The SMM can be used in various longitudinal studies with different monitoring techniques and/or biological systems. For example, analysis of the longitudinal blood flow measured by diffuse correlation technique could evaluate the healing in bone injury [16–18] or the efficacy of chemotherapeutic drugs in breast cancer [25]. The monitoring techniques could be any spectroscopy or imaging techniques as long as they can provide longitudinal data.

## Supporting information

**S1 Data.**
(CSV)

## Acknowledgments

The authors thank the Editor for handling the paper and providing constructive comments and suggestions.

## Author Contributions

**Conceptualization:** Regine Choe, Tong Tong Wu.

**Data curation:** Sami Leon, Jingxuan Ren.

**Formal analysis:** Sami Leon.

**Funding acquisition:** Regine Choe, Tong Tong Wu.

**Investigation:** Sami Leon, Jingxuan Ren, Regine Choe, Tong Tong Wu.

**Methodology:** Sami Leon, Tong Tong Wu.

**Project administration:** Tong Tong Wu.

**Resources:** Regine Choe, Tong Tong Wu.

**Software:** Sami Leon.

**Supervision:** Regine Choe, Tong Tong Wu.

**Validation:** Regine Choe, Tong Tong Wu.

**Visualization:** Sami Leon, Jingxuan Ren.

**Writing – original draft:** Sami Leon, Jingxuan Ren, Regine Choe, Tong Tong Wu.

**Writing – review & editing:** Sami Leon, Jingxuan Ren, Regine Choe, Tong Tong Wu.

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
