## [Editor Report · Decision Letter 0]

21 Dec 2021

PONE-D-21-28535Semiparametric Mixed-Effects Model for Analysis of Non-invasive Longitudinal Hemodynamic Responses During Bone Graft HealingPLOS ONE

Dear Dr. Tong Tong Wu,

Thank you for submitting your manuscript to PLOS ONE. After careful consideration, we feel that it has merit but does not fully meet PLOS ONE’s publication criteria as it currently stands. Therefore, we invite you to submit a revised version of the manuscript that addresses the points raised during the review process.

I’m not sure the authors proved that the suggested semiparametric mixed model is needed to answer the question if autograft group having higher total hemoglobin concentration than the allograft group, that is, there is a statistically significant treatment effect. The testing could be done in a much simpler and more flexible way by taking the difference between the two groups on each day and testing that the mean is zero. Either ANOVA or variance component/mixed model could applied. Note that the difference allows an arbitrary pattern of temporal changes of graft hemodynamic.

In addition, temporal changes exhibit much variation over time and I wonder if a complex statistical model such as SLMM has grounds for this specific application.

In summary, I doubt that SLMM is an appropriate model for detecting the treatment effect for these data but nevertheless I give the authors an opportunity to prove otherwise.

We look forward to receiving your revised manuscript.

Kind regards,

Eugene Demidenko, Ph.D.

Academic Editor

PLOS ONE

“RC and TTW’s work was partly supported by grant from the National Institutes of 206 Health NIH R01AR071363; TTW’s work was also supported by grant from the National 207 Science Foundation NSF CCF-1934962.”

“RC and TTW's work was partly supported by grant from the National Institutes of Health NIH R01AR071363; TTW's work was also supported by grant from the National Science Foundation NSF CCF-1934962. The funders had no role in study design, data collection and analysis, decision to publish, or preparation of the manuscript.”

3. Please include a caption for figure 2-7.

---

## [Author Response · Author response to Decision Letter 0]

28 Jan 2022

Dear Dr. Demidenko,

Thank you for handling our paper Semiparametric Mixed-Effects Model for Analysis of Non-invasive Longitudinal Hemodynamic Responses During Bone Graft Healing submitted to PLOS ONE. The comments and suggestions we received were very insightful. In submitting our revision, we have tried to address all the questions, which are highlighted in blue. We believe the revision has improved in quality and hope it meets your standard and can be quickly approved.

Sincerely,

Sami Leon

Jingxuan Ren 

Regine Choe

Tong Tong Wu

---

## [Decision Letter · Decision Letter 1]

28 Feb 2022

PONE-D-21-28535R1Semiparametric Mixed-Effects Model for Analysis of Non-invasive Longitudinal Hemodynamic Responses During Bone Graft HealingPLOS ONE

Dear Dr. Tong Wu,

Thank you for submitting your manuscript to PLOS ONE. After careful consideration, we feel that it has merit but does not fully meet PLOS ONE’s publication criteria as it currently stands. Therefore, we invite you to submit a revised version of the manuscript that addresses the points raised during the review process.

I believe one more round of revision is required: (1) although the reviewer was very positive overall his comments on the presentation are worth of consideration; (2) as I mentioned previously, semiparametric model is prone to overfitting by claiming the discovered nonlinear pattern as the pattern of noise. You have to address or at least mention this weakness in the revised version. Looking forward to receiving the updated version of your paper soon.

We look forward to receiving your revised manuscript.

Kind regards,

Eugene Demidenko, Ph.D.

Academic Editor

PLOS ONE

Journal Requirements:

Reviewers' comments:

Reviewer's Responses to Questions

**Comments to the Author**

1. If the authors have adequately addressed your comments raised in a previous round of review and you feel that this manuscript is now acceptable for publication, you may indicate that here to bypass the “Comments to the Author” section, enter your conflict of interest statement in the “Confidential to Editor” section, and submit your "Accept" recommendation.

Reviewer #1: All comments have been addressed

2. Is the manuscript technically sound, and do the data support the conclusions?

Reviewer #1: Yes

3. Has the statistical analysis been performed appropriately and rigorously? 

Reviewer #1: Yes

4. Have the authors made all data underlying the findings in their manuscript fully available?

Reviewer #1: Yes

5. Is the manuscript presented in an intelligible fashion and written in standard English?

Reviewer #1: Yes

6. Review Comments to the Author

Reviewer #1: I am happy with the revised version, however, the paper still needs some care. I recommend acceptance subject to making the following minor corrections:

1- The abbreviations "SMM" and "EM" are not needed in the Abstract nor the standing of LASSO

2- For the state of art in longitudinal data modeling, the authors can refer to Taavoni and Arashi (2021) and Taavoni et al. (2021).

3- The first time the abbreviation SMM is defined on page 2 is sufficient. There is no need to write the whole expression further on.

4- The first time the nonparametric component f is defined must be pointed that it belongs to the space of polynomials.

References

Taavoni, M. and Arashi, M. (2021) High-dimensional generalized semiparametric model for longitudinal data, Statistics, 55(4), 831-850.

Taavoni, M., Arashi, M., Wang, W.L. and Lin, T.I. (2021) Multivariate t semiparametric mixed-effects model for longitudinal data with multiple characteristics, Journal of Statistical Computation and Simulation, 92(2), 260-281.

7. PLOS authors have the option to publish the peer review history of their article (what does this mean?). If published, this will include your full peer review and any attached files.

Reviewer #1: No

---

## [Author Response · Author response to Decision Letter 1]

1 Mar 2022

Dear Dr. Demidenko,

Thank you for handling our revised paper Semiparametric Mixed-Effects Model for Analysis of Non-invasive Longitudinal Hemodynamic Responses During Bone Graft Healing. We have made all the suggested changes, including a short discussion on the overfitting issue. All the changes are highlighted in blue in the revision. We hope this revision meets your standard and can be quickly approved.

Sincerely,

Sami Leon

Jingxuan Ren 

Regine Choe

Tong Tong Wu

---

## [Editor Report · Decision Letter 2]

3 Mar 2022

Semiparametric Mixed-Effects Model for Analysis of Non-invasive Longitudinal Hemodynamic Responses During Bone Graft Healing

PONE-D-21-28535R2

Dear Dr. Wu,

We’re pleased to inform you that your manuscript has been judged scientifically suitable for publication and will be formally accepted for publication once it meets all outstanding technical requirements.

The authors addressed the comments of the Reviewers. The paper is acceptable now.

Kind regards,

Eugene Demidenko, Ph.D.

Academic Editor

PLOS ONE

---

## [Editor Report · Acceptance letter]

22 Mar 2022

PONE-D-21-28535R2 

Semiparametric Mixed-Effects Model for Analysis of Non-invasive Longitudinal Hemodynamic Responses During Bone Graft Healing 

Dear Dr. Wu:

I'm pleased to inform you that your manuscript has been deemed suitable for publication in PLOS ONE. Congratulations! Your manuscript is now with our production department. 

Kind regards, 

on behalf of

Dr. Eugene Demidenko 

Academic Editor

PLOS ONE